# Myc-regulated miRNAs modulate p53 expression and impact animal survival under nutrient deprivation

**María P. Gervé**[1], **Juan A. Sánchez**[1], **María C. Ingaramo**[1], **Andrés Dekanty**[1,2]*

1 Instituto de Agrobiotecnología del Litoral, Consejo Nacional de Investigaciones Científicas y Técnicas (CONICET), Santa Fe, Argentina, 2 Facultad de Bioquímica y Ciencias Biológicas, Universidad Nacional del Litoral (UNL), Santa Fe, Argentina

* adekanty@santafe-conicet.gov.ar

**Data Availability Statement:** All relevant data are within the manuscript and its Supporting Information files.

## Abstract

The conserved transcription factor Myc regulates cell growth, proliferation and apoptosis, and its deregulation has been associated with human pathologies. Although specific miRNAs have been identified as fundamental components of the Myc tumorigenic program, how Myc regulates miRNA biogenesis remains controversial. Here we showed that Myc functions as an important regulator of miRNA biogenesis in *Drosophila* by influencing both miRNA gene expression and processing. Through the analysis of ChIP-Seq datasets, we discovered that nearly 56% of *Drosophila* miRNA genes show dMyc binding, exhibiting either the canonical or non-canonical E-box sequences within the peak region. Consistently, reduction of dMyc levels resulted in widespread downregulation of miRNAs gene expression. dMyc also modulates miRNA processing and activity by controlling Drosha and AGO1 levels through direct transcriptional regulation. By using *in vivo* miRNA activity sensors we demonstrated that dMyc promotes miRNA-mediated silencing in different tissues, including the wing primordium and the fat body. We also showed that dMyc-dependent expression of miR-305 in the fat body modulates Dmp53 levels depending on nutrient availability, having a profound impact on the ability of the organism to respond to nutrient stress. Indeed, dMyc depletion in the fat body resulted in extended survival to nutrient deprivation which was reverted by expression of either miR-305 or a dominant negative version of Dmp53. Our study reveals a previously unrecognized function of dMyc as an important regulator of miRNA biogenesis and suggests that Myc-dependent expression of specific miRNAs may have important tissue-specific functions.

## Author summary

Myc is a conserved transcription factor that regulates the expression of a large number of genes, and has long been recognized as a major driver of human tumorigenesis. In addition to well-described Myc target genes such as those involved in ribosome biogenesis, Myc is also able to regulate the expression of microRNAs (miRNAs). Both Myc and miR-NAs have been shown to influence cell proliferation, and their deregulation has been

**Funding:** M.P.G., M.C.I and J.A.S are funded by PhD fellowships from CONICET. A.D. is a member of CONICET and Professor at UNL. This work was supported by the Agencia Nacional de Promoción Científica y Tecnológica, Argentina (PICT-2017-2036 and PICT-2019-01319 to A.D.). The funders had no role in study design, data collection and analysis, decision to publish, or preparation of the manuscript.

**Competing interests:** The authors have declared that no competing interests exist.

associated with different human pathologies. Indeed, specific miRNAs have been identified as fundamental components of the Myc tumorigenic program. In this work, we find that *Drosophila* Myc (dMyc) regulates the expression of a large number of miRNAs through direct transcriptional regulation. We also find that Myc-dependent expression of specific miRNAs may have important tissue-specific functions. Through the regulation of miR305 in the adipose tissue, dMyc maintain low p53 levels under normal conditions. In contrast, under nutrient stress, reduced dMyc levels alleviates miR305-mediated repression of p53. Once activated, p53 promotes metabolic homeostasis and organismal survival to nutrient stress. Given the role of dMyc and miRNAs in human pathologies, our findings will be important in understanding the complex regulatory mechanisms of physiological and pathological responses in humans.

## Introduction

Myc is a conserved basic helix–loop–helix leucine zipper (bHLHZ) transcription factor that regulates the expression of an estimated 10–15% of human genes [1]. This transcription factor has long been recognized as a key regulator of cell growth, proliferation and metabolism and its deregulation has been associated with human cancer [2,3]. Myc forms a heterodimeric DNA binding complex with the small bHLHZ protein Max and binds to the canonical enhancer-box (E-box) sequence CACGTG to modulate transcription of nearby genes [1,3,4]. The number of Myc-regulated genes is large, and encompasses a broad range of biological functions including virtually every anabolic and growth-promoting process [3].

Emerging evidence has demonstrated that Myc is also able to regulate the expression of non-coding RNAs [2,5–8]. MicroRNAs (miRNAs) are short non-coding RNAs that regulate the expression of target genes by directly binding to their mRNAs [9]. Hundreds of miRNAs have been listed in higher animals, and deregulation of many of them has been associated with cancer [10]. The biogenesis of miRNA is tightly regulated at both transcriptional and post-transcriptional levels. Primary miRNAs (pri-miRNAs) are transcribed by RNA Pol II and folded into a hairpin-like structure, facilitated by the presence of complementary sequences. The hairpin structure of the pri-miRNA is recognized and processed by an enzyme called Drosha, resulting in the production of a shorter RNA molecule called a precursor miRNA (pre-miRNA). The pre-miRNA is then exported from the nucleus to the cytoplasm, where it undergoes further processing by Dicer-1 (Dcr-1), leading to the production of $\sim$22-nt miRNA duplexes. Once loaded into the Argonaute-1 (AGO-1) containing miRNA-Induced Silencing Complex (miRiSC), the mature single-stranded miRNA guides the miRISC to the 3′ untranslated region (UTR) of target mRNAs [11–13].

Transcription of miRNAs is regulated by RNA Pol II-associated transcription factors (TFs) and epigenetic regulators [9]. Among these TFs, Myc has been shown to either activate or repress transcription of specific miRNA genes. On one hand, Myc has been shown to activate expression of the miR-17–92 cluster, a pro-tumorigenic group of six miRNAs, in human cells [7,8]. Expression of the miR-17–92 cluster is frequently observed in tumors, promotes proliferation in cell lines, and accelerates tumorigenesis in mouse models of Myc-induced colon cancer and lymphoma [7,8]. On the other hand, a large set of additional miRNAs have been shown to be repressed by Myc in human and mouse models of B cell lymphoma [6]. Interestingly, expression of Myc-repressed miRNAs markedly impairs lymphoma cell growth in vivo [6]. Myc has also been shown to modulate miRNA processing via the transcriptional regulation of Drosha in P493-6 human B cell line [14]. Although not entirely understood, these

observations point towards Myc as an important regulator of miRNA biogenesis and suggest that Myc-dependent activation and/or repression of specific miRNAs is a fundamental component of the Myc tumorigenic program.

Similarly to its human counterpart, the single *Drosophila* Myc protein (dMyc) also plays an essential role in controlling cell growth and proliferation through the transcriptional regulation of a large number of target genes [15–21]. Although dMyc has been shown to regulate the expression of specific miRNAs [22,23], the exact role of dMyc in regulating miRNA expression and/or processing is largely unknown. In this study, we identified a large set of Myc-regulated miRNAs in *Drosophila*. We showed that dMyc regulates miRNA transcription by directly binding to either the canonical or non-canonical E-box sequences located in at least 56% of the *Drosophila* miRNAs. In addition, direct transcriptional regulation of Drosha and AGO1 by dMyc also allows for modulation of miRNA processing and activity. Through *in vivo* miRNA activity sensors, we demonstrate that dMyc promotes miRNA-mediated silencing in different *Drosophila* tissues such as the wing primordium and the fat body, a functional analog of vertebrate liver and adipose tissue. Interestingly, dMyc-dependent expression of miR-305 in the fat body is critical to modulate Dmp53 levels (the *Drosophila* ortholog of mammalian p53), this being crucial for metabolic homeostasis maintenance and animal survival upon nutrient stress. Our study reveals a previously unrecognized function of dMyc as a major regulator of miRNA expression and processing.

## Results

### Myc regulation of miRNAs gene expression in *Drosophila*

To explore a possible role of dMyc in regulating miRNA gene expression we made use of the gene-switch GAL4 system (GSG) that allows for conditional expression of a $myc^{RNAi}$ transgene [24]. An inactive GAL4-progesterone-receptor fusion protein is expressed under the control of an ubiquitous promoter and becomes active only in the presence of the activator RU-486 [24]. As shown in Fig 1A, $GSG^{162} > myc^{RNAi}$ animals exposed to RU-486 for 72 h showed reduced *dmyc* transcript levels when compared to vehicle-treated controls. We then measured the levels of primary miRNA (pri-miRNA) for a randomly selected group of miRNA genes by qRT-PCR and observed that expression of almost every tested pri-miRNAs was downregulated in $GSG^{162} > myc^{RNAi}$ animals (Fig 1B). Interestingly, $GSG^{162} > myc^{RNAi}$ animals also presented reduced levels of precursor miRNAs (pre-miRNA) and mature miR-219 and miR-305 (Fig 1C and 1D). No differences were observed in the levels of miR-8 and miR-283 (Fig 1C), most probably indicating differences in the stability and/or turnover of mature miRNAs. We then evaluated the effect of overexpressing dMyc on pri-miRNA expression by using a transgenic line expressing dMyc under the control of a *tubulin* promoter (*tub*>dMyc; [18]) which resulted in a two to three-fold increase in *dmyc* transcript levels (S1A Fig). In this case, we observed the upregulation of pri-miR-283, pri-miR-306, pri-miR-6, and pri-miR-2a, while no significant differences were detected for other pri-miRNAs (S1B Fig). The absence of significant effects on miRNA transcription upon dMyc overexpression may be attributed to the saturation of dMyc occupancy on the promoter regions of miRNA genes. Collectively, these findings provide compelling evidence for the regulatory role of dMyc in controlling the expression of a substantial number of miRNAs in *Drosophila*.

### dMyc association to miRNA genomic region

A curated database of dMyc-binding sites obtained from whole-genome Chromatin IP sequencing (ChIP-Seq) assays using whole L3 *Drosophila* larvae have been made available by the modENCODE Consortium [25]. We therefore used modENCODE datasets to evaluate

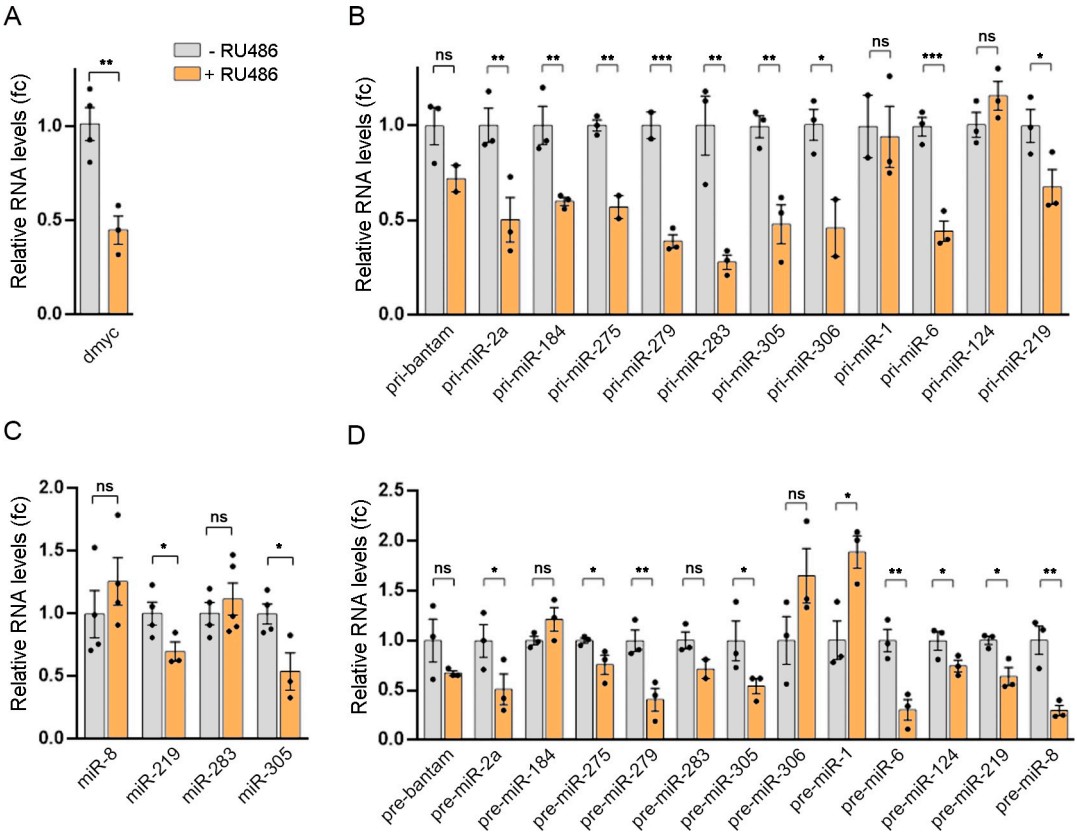

**Fig 1. Myc-mediated regulation of miRNA gene expression in Drosophila.** (A) qRT-PCR showing *myc* transcript levels in GSG$^{162}$>*myc$^{RNAi}$* larvae in the presence of RU486 or vehicle. (B-D) qRT-PCR showing relative RNA levels of the indicated primary (B), precursor (D) or mature (C) miRNAs in GSG$^{162}$>*myc$^{RNAi}$* larvae in the absence or presence of RU486. Results are expressed as fold induction with respect to control animals. Three independent replicates were carried out for each sample. Mean ± SEM. Unpaired two-tailed t-tests: * p<0.05; ** p<0.01; ***p<0.001; ns: not significant. See also S1 Fig.

whether miRNA genes are direct targets of dMyc (http://epic.gs.washington.edu/modERN). We focused the analysis on the 253 miRNA genes listed in the *Drosophila* miRNA database (miRBase Release 22.1, http://www.mirbase.org), and found 111 dMyc regulatory regions potentially covering 119 miRNAs (Fig 2A and S1 Table, see Materials and methods for details). Similar results were obtained when using dMyc ChIP-Seq datasets from *Drosophila* cultured cells available in the TransmiR v2.0 database [26] (Fig 2A). Interestingly, a total of 142 miRNAs were identified in at least one of these datasets, representing 56% of annotated miRNA transcripts in the *Drosophila* genome. Further characterization of the dMyc-associated miRNAs genomic regions showed a high presence of either canonical or non-canonical E-box sequences previously described for human Myc proteins, suggesting a direct role of dMyc in the regulation of miRNAs expression (Figs 2B, 2C and S2 and S1 Table, see Materials and methods for details). An enrichment analysis based on the hypergeometric distribution revealed a significant over-enrichment of miRNA genes, with a fold enrichment of 2.64 compared to random expectations (hypergeometric p-value = 7.7e-25). This finding strongly supports the highly significant association between dMyc and miRNA genes.

We confirmed dMyc binding to the genomic locus of a selected group of miRNA genes by performing ChIP-qPCR experiments using a previously validated anti-dMyc antibody [27]. We observed significant enrichment of the selected miRNA genes and the positive control

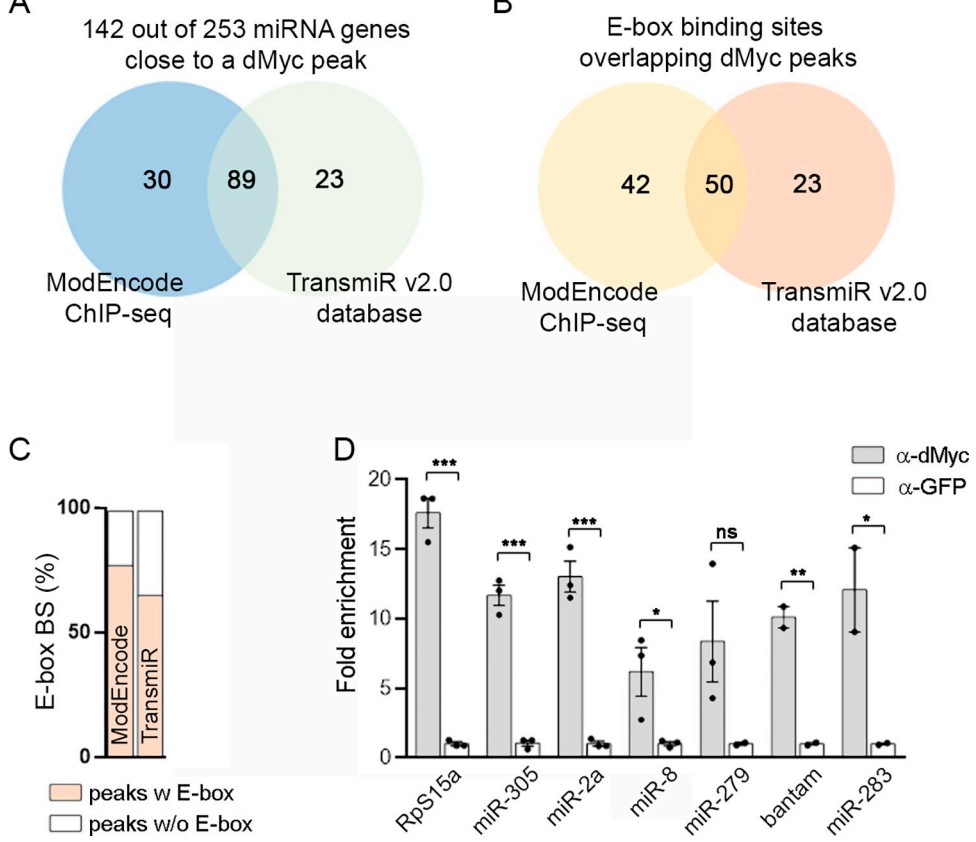

**Fig 2. Identification of Myc-regulated miRNAs in *Drosophila*.** (A) Venn diagrams showing overlap between ChIP-seq databases for dMyc [25,26]. (B-C) Canonical and non-canonical E-box sequences overlapping dMyc peaks in the locus of candidate dMyc-regulated miRNA genes. (D) ChIP-qPCR assays showing dMyc binding to the locus of indicated genes. Results are expressed as fold enrichment with respect to control anti-GFP samples. Mean ± SEM. Unpaired two-tailed t-tests: * p<0.05; ** p<0.01; ***p<0.001; ns: not significant. See also S1 Table and S2 Fig.

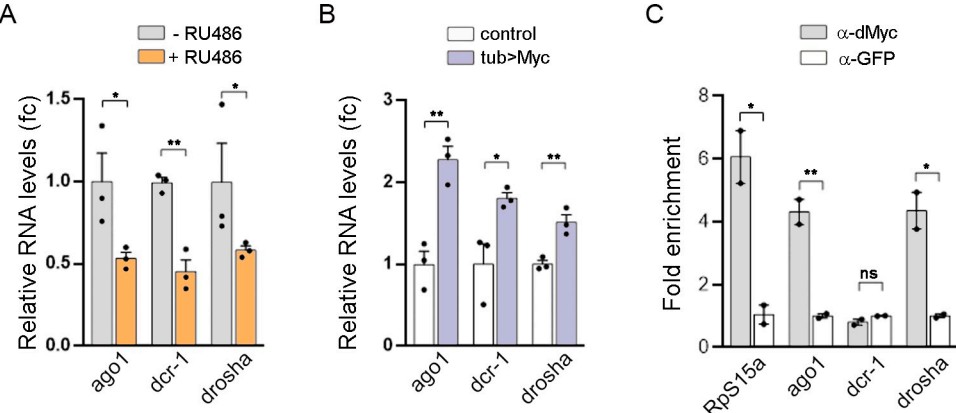

**Fig 3. Myc regulation of the miRNA machinery.** (A-B) qRT-PCR showing *dcr1*, *drosha* and *ago1* mRNA levels in GSG[162]>*myc*[RNAi] larvae treated with RU486 or vehicle (A), as well as in *tub*>dMyc larvae (B). Results are expressed as fold induction relative to control animals. Three independent replicates were carried out for each sample. (C) ChIP-qPCR assays showing dMyc binding to the promoter region of the indicated genes. Results expressed as fold enrichment with respect to control anti-GFP samples. Mean ± SEM. Unpaired two-tailed t-tests: * p<0.05; ** p<0.01; ***p<0.001; ns: not significant.

RpS15a [27] in dMyc-ChIP samples when compared with the negative control (Fig 2D). Altogether, these findings demonstrate *in vivo* association of dMyc with immediate regions next to 115 miRNA genes that contain either the canonical or the non-canonical E-box sequences, providing strong evidence that these miRNAs are directly regulated by dMyc.

## dMyc regulation of main miRNA biogenesis factors

Emerging data have provided evidence that Myc is able to regulate the expression of Drosha, an enzyme involved in the nuclear processing of pri-miRNAs [14]. Interestingly, the transcript levels of *dcr-1*, *drosha* and *ago1* were strongly reduced in larvae expressing *myc^RNAi* (Fig 3A), whereas they were upregulated in *tub>dMyc* animals (Fig 3B). We then evaluated dMyc binding to the genomic locus of these genes by ChIP-qPCR and observed a significant association of dMyc to the promoter region of *ago1* and *drosha* (Fig 3C). However, dMyc was not found to be enriched in *dcr-1* promoter, meaning the effect of dMyc depletion on Dcr-1 levels might be indirect. Taken together, these results suggest that dMyc activity is required for miRNA biogenesis to regulate the expression of both pri-miRNAs and central elements of the miRNA machinery.

## Myc regulates miRNA activity sensor in the *Drosophila* wing primordium

The expression of the *Drosophila* ortholog of mammalian p53 (Dmp53) has previously been shown to be regulated by miRNAs in both the wing imaginal disc and the fat body [28]. Notably, three specific miRNAs (miR-283, miR-219, miR-305), previously identified as regulators of Dmp53 expression in the wing primordium [28], showed reduced levels upon *myc^RNAi* expression (Fig 1B) and exhibited dMyc binding to its genomic region (Fig 2D and S1 Table). We then made use of a miRNA sensor that exploits the regulation of p53 expression by miRNAs in the larval wing primordium to characterize dMyc contribution in miRNA biogenesis. To assess for miRNA-mediated regulation of Dmp53 we used a *p53-3'UTR* sensor consisting of the wild type *Dmp53 3'UTR* cloned into a *tubulin*-promoter-EGFP reporter plasmid, showing increased GFP expression levels when interfering with the miRNA machinery (S3 Fig; [28]). Consistent with a role of dMyc in regulating miRNA biogenesis, expression of *myc^RNAi* in a stripe of anterior cells adjacent to the AP boundary, using *ptc*-Gal4, led to increased GFP levels of the *p53*-sensor (Fig 4A and 4B, see also Fig 4D for quantifications of GFP intensity levels). Conversely, dMyc overexpression using the *dpp*-Gal4 driver resulted in a significant reduction in the GFP signal (Fig 4C and 4D). It is noteworthy that a previous report indicated that dMyc overexpression in the wing primordium leads to an increase in the activity levels of bantam [29]. We then extended our analysis to include alternative miR-GFP sensors based on Mei-p26 or dMyc 3'UTRs, known to be regulated by miRNAs in the *Drosophila* wing imaginal disc [30,31]. Intriguingly, dMyc-depletion increased the expression of these miR-GFP sensors (Fig 4D, 4E and 4G) and dMyc overexpression resulted in a strong reduction of GFP levels (Fig 4D, 4F and 4H) compared with control discs showing ubiquitous GFP expression (S3 Fig). These findings indicate that the regulatory influence of dMyc extends beyond miRNAs targeting Dmp53. Moreover, our results indicate that dMyc plays a crucial role in regulating miRNA biogenesis in the *Drosophila* wing primordium, influencing either miRNA gene expression or processing.

## dMyc regulates miR-305 expression in the fat body according to nutrient availability

Expression of core elements of the miRNA machinery have been previously shown to be regulated by nutrient availability in the FB of starved animals [28]. Upon starvation, reduced Dcr-1 levels and impaired miRNA biogenesis results in Dmp53 activation which, in turn, promotes metabolic homeostasis and organismal survival to nutrient stress [28,32]. Accordingly, mRNA

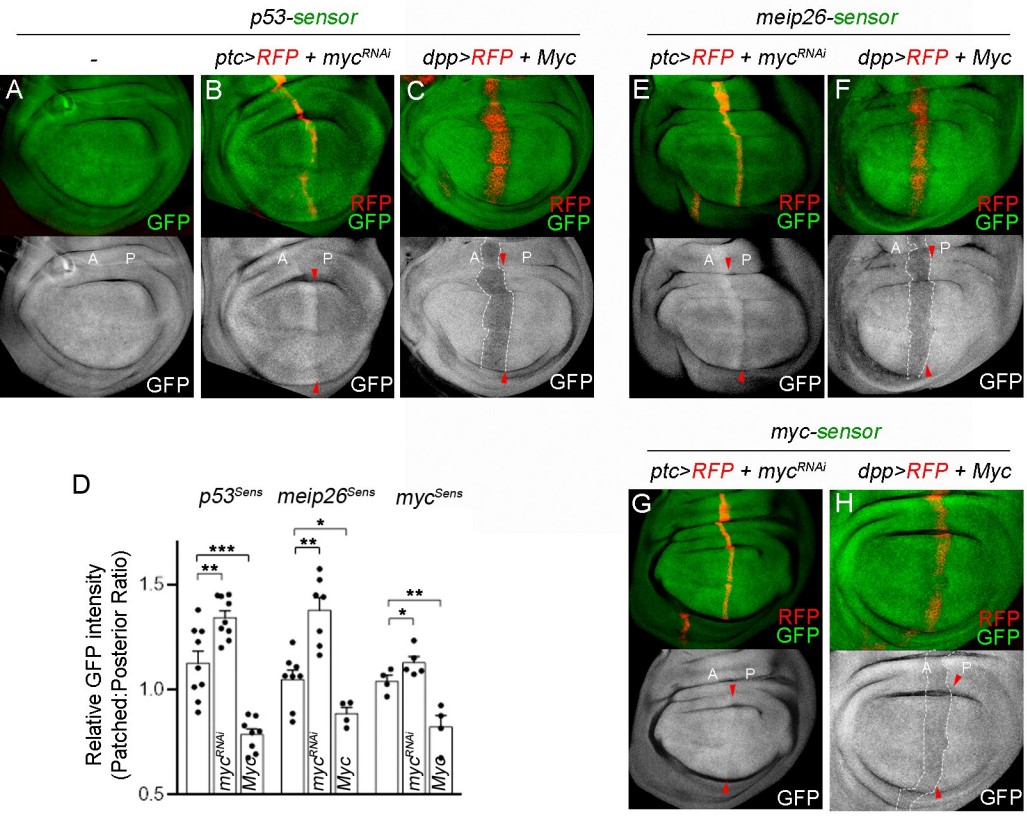

**Fig 4. Myc regulation of miRNA activity sensor in the wing primordium.** (A-C) Wing discs carrying the p53-sensor along with the indicated transgenes and stained to visualize GFP (in green or white). (D) Relative GFP intensity levels between Anterior (patched) and Posterior domains of wing discs from the indicated genotypes. (E-H) Wing discs carrying the meip26-sensor (E-F) or myc-sensor (G-H) along with the indicated transgenes and stained to visualize GFP (in green or white). The wing discs in (B,E, G) expressed *myc^{RNAi}* under the control of the *ptc*-Gal4 driver (marked by the expression of RFP, in red). The wing discs in (C,F, H) expressed dMyc under the control of the *dpp*-Gal4 driver (marked by the expression of RFP, in red). Note increased GFP levels in dMyc-depleted cells, but reduced expression in dMyc-overexpressing cells. Red arrowheads depict the anterior-posterior (A-P) boundary. A: anterior; P: posterior. See also S3 Fig.

levels of *dcr-1* and *drosha* were significantly reduced after 24h starvation treatment in both adult abdomens [28] and larval FB (Fig 5A). dMyc transcript levels were also reduced in the FB of starved animals (Fig 5A) and expression levels of *dcr-1* and *drosha* were strongly reduced in larvae expressing *myc^{RNAi}* (Fig 3A), thus raising the possibility that decreased expression of miRNA biogenesis genes in the FB might be a direct consequence of reduced dMyc levels. However, even though *tub*>dMyc animals showed sustained dMyc levels after 24h starvation, ubiquitous expression of dMyc was not able to rescue the reduced levels of *dcr-1* and *drosha* observed in the FB of starved animals (Fig 5C). Similar results were obtained by overexpressing dMyc with the GSG^{162}-Gal4 driver in the presence of RU486 (S4 Fig). Among the three miR-NAs previously shown to regulate Dmp53 expression in the wing primordium only miR-305 has been shown to repress Dmp53 expression in the FB [28]. Interestingly, expression of pri-miR-305 was significantly reduced in the FB of dMyc-depleted animals (Fig 5B) and showed dMyc binding to its promoter region (Fig 2A and 2D), raising the possibility that miR-305 expression itself might be compromised under nutrient deprivation. Indeed, pri-miR-305 levels were reduced in the FB of starved larvae and dMyc overexpression was sufficient to revert

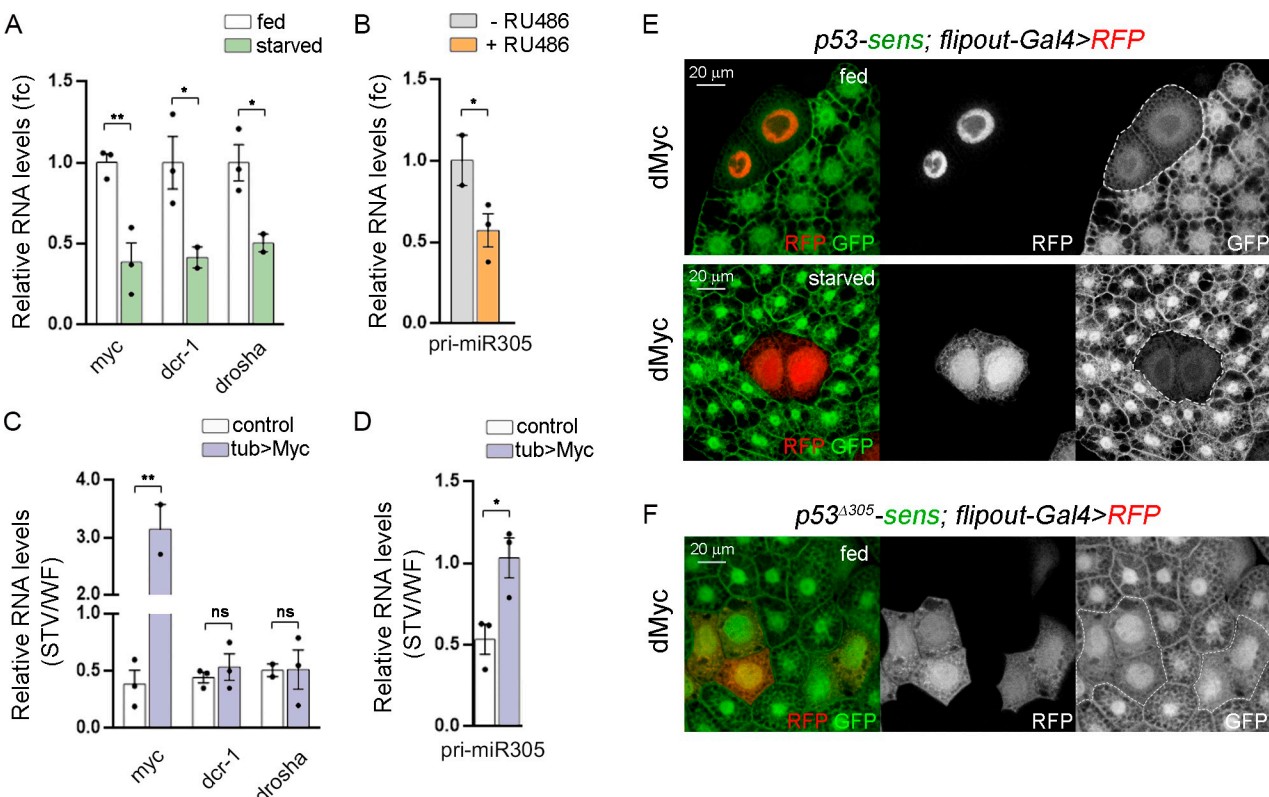

**Fig 5. dMyc-dependent regulation of pri-miR-305 in the fat body.** (A) qRT-PCR showing *myc*, *dcr1* and *drosha* mRNA levels in the FB of larvae subjected to fed or starved conditions. (B) qRT-PCR showing pri-miR-305 levels in the FB of GSG$^{162}$>*myc$^{RNAi}$* larvae in the absence or presence of RU486. (C-D) qRT-PCR showing *myc*, *dcr1* and *drosha* transcript levels (C) or pri-miR-305 levels (D) in the FB of larvae from the indicated genotypes subjected to fed (WF) or starved (STV) conditions. Results are expressed as fold induction with respect to control animals. (E-F) The flipout-Gal4 line was used to overexpress dMyc in single fat body cells, allowing us to investigate the influence of dMyc expression on p53-sensor levels. Notably, expression of dMyc in single fat body cells (marked by RFP, in red or white) led to a decrease in p53-sensor levels of both fed and starved animals (E; in green or white), while having no effect on the expression of the p53Δ305-sensor (F; in green or white). Mean ± SEM. Unpaired two-tailed t-tests: * p<0.05; ** p<0.01; ***p<0.001; ns: not significant. Scale bars, 20 µm. See also S4 Fig.

pri-miR-305 expression to normal levels (Fig 5D). Together, these findings suggest that the transcription of miR-305 is directly regulated by dMyc, while the expression of Dcr-1 and Drosha in the fat body under nutrient deprivation involves a dMyc-independent mechanism. Interestingly, a recent study has reported that the transcription factor FOXO represses Dcr-1 expression levels under nutrient deprivation, indicating the involvement of alternative regulatory pathways [33]. Further investigation is required to determine whether Drosha is also negatively regulated under nutrient stress.

## dMyc-dependent regulation of miR-305 expression contributes to starvation resistance

We then decided to study the functional consequences of dMyc-dependent expression of miR-305 on Dmp53 levels and organismal survival upon nutrient stress. To monitor Dmp53 repression by miR-305, we used either *p53-3'UTR* sensor or *p53$^{Δ305}$–3'UTR* sensor lacking the predicted miR-305 binding site [28]. When exposed to nutrient deprivation, the *p53*-sensor showed increased GFP levels while the *p53$^{Δ305}$*-sensor showed no changes in this parameter [28]. Consistent with a role of dMyc in controlling miR-305-mediated repression of Dmp53, dMyc overexpressing cells showed reduced *p53*-sensor levels in fat body cells of both well fed

and starved animals, but had no effect on the expression of the $p53^{\Delta305}$-sensor (Figs 5E, 5F and S4; Myc expressing cells were marked by the expression of red fluorescent protein).

Next, we evaluated whether dMyc-mediated expression of miR-305 in the FB has an impact on the survival rates of adult flies subjected to nutrient deprivation and whether this effect relied on the activity of Dmp53. Targeted expression of $myc^{RNAi}$ in FB cells ($Lsp2>myc^{RNAi}$ flies) increased the survival rates of both male and female adult flies subjected to nutrient starvation when compared to control flies (Fig 6A and 6B and S2 Table). Interestingly, the extended survival rates were reverted upon coexpression of either a miR-305 hairpin [34]

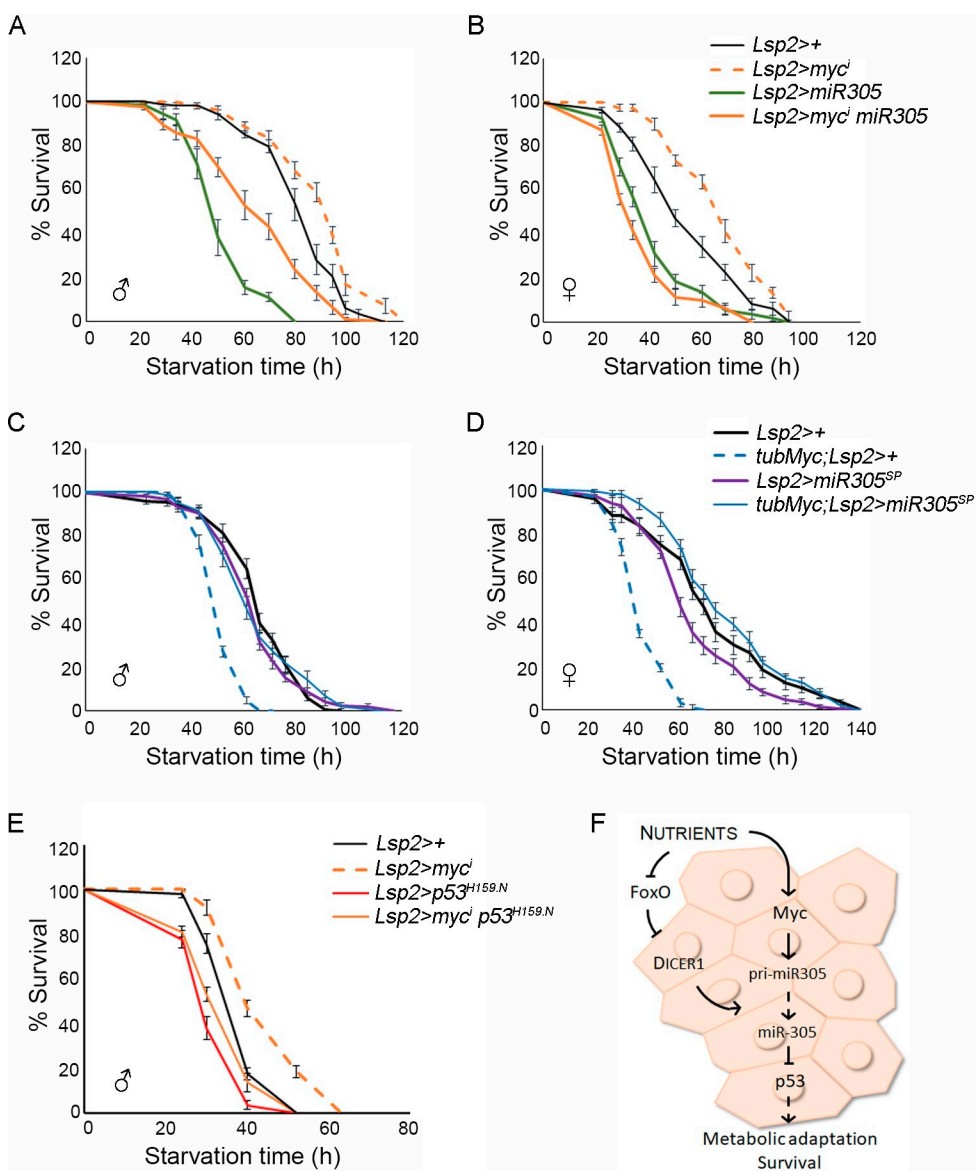

**Fig 6. Starvation Resistance upon FB-Specific modulation of dMyc, miR-305 and Dmp53.** (A-F) Survival rates to nutrient deprivation of adult flies of the indicated genotypes compared to control flies subjected to the same procedure. Both male (A, C, E) and female (B, D) adult flies were analyzed. See S2 Table for n, p-value, median, and maximum survival values. Error bars represent SEM. (G) dMyc stimulates the transcription of pri-miR305, leading to the repression of Dmp53 expression in the FB of fed animals. Under nutrient deprivation, dMyc levels decrease, and FOXO is activated, impairing miR-305 expression and processing. As a consequence, Dmp53 is activated, which facilitates energy homeostasis and supports animal survival during nutrient stress. See also S5 Fig.

(*Lsp2>myc^RNAi*, *miR-305* flies; Fig 6A and 6B and S2 Table) or the dominant-negative form Dmp53^H159.N (*Lsp2>myc^RNAi*, *p53^H159.N* flies; Figs 6E, S5C and S2 Table). Similar results were obtained when modulating dMyc or miR-305 levels specifically in the adult FB using the GSG^132-Gal4 driver in the presence of RU486 (S5A Fig and S2 Table). Conversely, whereas ubiquitous dMyc expression (*tub>*dMyc flies) compromised the survival rates of adult flies exposed to starvation conditions (Fig 6C and 6D and S2 Table), reducing miR-305 activity by expression of a miR-305 sponge construct [35] in the FB completely reverted the starvation sensitivity caused by dMyc overexpression (*tubMyc; Lsp2>miR-305^sponge* flies; Fig 6C and 6D and S2 Table). Finally, in accordance with a model where Dmp53 activation enhances survival, the conditional expression of Dmp53 in the FB of adult flies increased survival rates during nutrient deprivation (S5B Fig). Altogether, these findings indicate that dMyc regulates miR-305 expression in the FB in a nutrient dependent manner, therefore influencing Dmp53 levels and the resistance to nutrient stress.

## Discussion

Although c-Myc-dependent regulation of specific miRNAs has been positioned as a fundamental component of the c-Myc tumorigenic program, how c-Myc modulates miRNA biogenesis remains controversial. In this study, we provide evidence that dMyc functions in *Drosophila* as an important regulator of miRNA biogenesis, influencing both miRNA gene expression and processing. By combining bioinformatic and experimental data we demonstrate that dMyc associates to the locus of a large set of miRNA genes, and reduction of dMyc levels results in widespread downregulation of miRNA expression. Eleven such miRNAs have previously been identified as putative dMyc-regulated miRNAs based on *in silico* promoter analysis [22], and two of them (miR-277 and miR-308) have been validated as dMyc-targets [22,23]. We also conclude that dMyc can affect miRNA processing and activity by controlling *drosha* and *ago1* levels through direct transcriptional regulation. Previous studies in mice and human cell lines have indicated that c-Myc is able to either activate or repress transcription of specific miRNA genes, as well as to modulate the processing of miRNAs via transcriptional regulation of *drosha*, thus suggesting a conserved mechanism [6–8,14]. Even though our results suggest that dMyc positively regulates miRNA gene expression, we cannot rule out the possibility of an opposite effect of dMyc on miRNA transcription depending on cell type and physiological context.

By using *in vivo* miRNA activity sensors we demonstrate that dMyc promotes miRNA-mediated silencing in different tissues, including the wing primordium and fat body. Interestingly, dMyc-dependent expression of miR-305 in the fat body modulates Dmp53 levels according to nutrient availability, which has a profound impact on the ability of the organism to respond to nutrient stress. dMyc depletion in the fat body resulted in extended survival to nutrient deprivation, which was reverted by coexpression of either miR-305 or a dominant negative version of Dmp53. Myc has also been shown to modulate the *Drosophila* immune response via direct activation of miR-277 transcription [22], strongly suggesting that Myc-dependent expression of specific miRNAs may have important tissue-specific functions. Along with its role in the regulation of Dmp53 levels in the adipose tissue, miR-305 has been shown to mediate adaptive homeostasis in the gut [36] and to participate in the aging process [37]. Interestingly, miR-305 expression in intestinal stem cells is under nutritional control [36]. As an age-related miRNA, ubiquitous expression of *miR-305* promotes the accumulation of protein aggregates in the muscle, leading to age-dependent impairment in locomotor activity, and reduced lifespan [37]. Conversely, miR-305 depletion in *Drosophila* adults suppresses these phenotypes and extended lifespan [37]. Together, these findings provide clear evidence that

miR-305 is required in different tissues for adaptive homeostasis in response to changing environmental conditions. To what extent dMyc-dependent regulation of miR-305 expression has an impact on gut homeostasis and longevity remains to be studied.

Dcr-1 has been shown to play a critical role in regulating metabolism and stress resistance in flies [28], as well as to promote cell cycle progression during *Drosophila* wing development [30]. Importantly, Dcr-1 has been identified in the adipose tissue of both mice and *Drosophila* as a rate-limiting enzyme in the biogenesis of miRNAs [28,38]. Under nutrient deprivation, the processing of miRNAs is impaired as a consequence of reduced Dcr-1 expression [28]. We examined Dcr-1 levels in response to dMyc expression and found reduced *dcr-1* mRNA levels in dMyc-depleted animals. However, our ChIP assay showed no association of dMyc to the Dcr-1 locus suggesting that the effect of dMyc depletion on Dcr-1 levels might be indirect. Most importantly, whereas dMyc overexpression reverted the reduced expression of pri-miR-305 under starvation, it was not able to rescue the reduced levels of *dcr-1* observed in the FB of starved animals. These results indicate that whereas miR-305 transcription requires direct action of dMyc, a dMyc-independent mechanism is involved in the regulation of Dcr-1 expression under nutrient deprivation. Accordingly, the transcription factor FOXO has recently been shown to repress Dcr-1 expression in the *Drosophila* fat body through direct transcriptional regulation [33]. JNK-dependent activation of FOXO in the fat body of starved animals represses Dcr-1 expression which, in turn, leads to reduced miRNA biogenesis. In this scenario, dMyc promotes the transcription of pri-miR305, resulting in the repression of Dmp53 expression in the fat body of well-fed animals (Fig 6F). Under nutrient deprivation, dMyc levels decrease, and FOXO becomes activated, impairing miR-305 expression and processing. This, in turn, leads to the activation of Dmp53, promoting energy homeostasis and enhancing animal survival under nutrient stress (Fig 6F). Altogether, these findings illustrate how different transcription factors converge on regulating miRNA biogenesis in a tissue-specific and context-dependent manner.

## Materials and methods

### *Drosophila* strains and maintenance

The following *Drosophila* strains were used: GSG[162] (BL40260), GSG[132] (BL8527), Lsp2-Gal4 (BL6357), Cg-Gal4 (BDSC:7011); UAS-myc[RNAi] (VDRC2947); UAS-dMyc (BL9674); tub-dMyc [18]; UAS-Dmp53[H159N] (BL8420); UAS-Dmp53 (BL6584); UAS-miR-305 (BL41152); UAS-miR-305[sponge] (BL61423); p53[3'UTR]-sensor [28]; p53Δ305[3'UTR]-sensor [28]; Myc[3'UTR]-sensor [31]; meiP26[3'UTR]-sensor [30]. Other stocks are described in Flybase. Flies were reared at 25˚C on standard media containing: 4% glucose, 40 g/L powder yeast, 1% agar, 25 g/L wheat flour, 25 g/L cornflour, 4 ml/L propionic acid and 1.1 g/L nipagin. For experiments using the Gene Switch system, embryos containing the GS-Gal4 driver and the UAS-myc[RNAi] transgene were collected over 24 h and maintained in regular food. In order to induce transgene expression, second instar larvae were transferred to food containing 50 μg/ml of RU486 (Sigma) or an equivalent volume of MilliQ water as a control. After 72h, full larvae or dissected tissues were used for RNA extraction.

### Fly husbandry and mosaic analysis

The Gal4/UAS binary system was used to drive transgene expression in the different *Drosophila* tissues [39] and experimental crosses were performed at 25˚C, unless otherwise specified. Crossing all Gal4 driver lines to the $w^{1118}$ background provided controls for each experiment. The Flp/Out system was used to generate RFP-marked clones. Flies from *hsFLP; act>y +>Gal4, UAS-RFP* were crossed to corresponding UAS-transgene lines at 25˚C and

spontaneous recombination events taking place in the fat body prior to the onset of endoreplication were analyzed [40].

## Starvation treatments and survival experiments

For starvation treatments in larvae, eggs were collected for 4 h intervals and larvae were transferred to vials containing standard food immediately after eclosion (first instar larvae, L1) at a density of 50 larvae per tube. Larvae were then raised at 25˚C for 72h prior to the starvation assay. Mid-third instar larvae were washed with PBS and placed in inverted 60 mm petri dishes with phosphate-buffered saline (PBS) soaked Whatman paper (starvation, STV) or maintained in standard food (well fed, WF). Each plate was sealed with parafilm and incubated at 25˚C for the duration of the experiment. After the starvation period, full larvae or dissected fat bodies were used for immunostaining or RNA extraction. For starvation sensitivity assays, 5- to 7-day-old flies of each genotype were transferred into vials containing 2% agar in PBS. Flies were transferred to new tubes every day, and dead flies were counted every six hours. Control animals were always analyzed in parallel in each experimental condition. Number of individuals used in each experiment is detailed in S2 Table. For starvation sensitivity assays using the GSG[132]-Gal4 line (S5 Fig), newly eclosed adults were transferred to food supplemented with 50 μg/ml of RU486 (Sigma). After 5- to 7-days, flies of each genotype were transferred into vials containing 2% agar in PBS along with 50 μg/ml of RU486.

## Immunostainings

Mid-third instar larvae were dissected in cold PBS and fixed in 4% formaldehyde/PBS for 20 min at room temperature. They were then washed and permeabilized in PBT (0.2% Triton X-100 in PBS) for 30 min and blocked in BBT (0.3% BSA, 250 mM NaCl in PBT) for 1 h. Samples were incubated overnight at 4˚C with primary antibody diluted in BBT, washed three times (15 min each) in BBT and incubated with secondary antibodies for 1.5 hours at room temperature. After three washes with PBT (15 min each), dissected tissues were placed in a mounting medium (80% glycerol/PBS containing 0.05% n-Propyl-Gallate). The following primary antibodies were used: mouse anti-GFP (12A6, DSHB); mouse anti-dMyc (P4C4-B10, DSHB); mouse anti-Dmp53 (7A4, DSHB). The following secondary antibodies were used: anti-mouse IgG-Alexa Fluor 488; anti-mouse IgG-Alexa Fluor 594 (Jackson InmunoResearch).

For fluorescence quantification, confocal images were taken using a Leica SP8 confocal microscope maintaining identical microscope settings for both control and experimental samples. Using Fiji software, fluorescence intensity were measured within defined areas of the Anterior (patched) and Posterior domains and represented as an A/P ratio. At least 10 images from independent experiments were analyzed per genotype and condition. Student's t test was used for statistical analysis.

## RNA isolation and quantitative RT-PCR

To measure transcript levels, total RNA was extracted from whole larvae or dissected FBs of 30 animals using TRIZOL RNA Isolation Reagent (Invitrogen). First strand cDNA synthesis was performed using an oligo(dT)18 primer and RevertAid reverse transcriptase (ThermoFisher) under standard conditions. Quantitative PCR was performed on an aliquot of the cDNA with specific primers using the StepOnePlus Real-Time PCR System. Expression values were normalized to *actin* transcript levels. In all cases, three independent samples were collected from each condition and genotype, and duplicate measurements were taken. Student's t test was used for statistical analysis. In the case of pre-miRNAs, first strand cDNA synthesis was performed using random hexamers. To quantify mature miRNA levels, we followed a two-step

process [41]: (1) RT with a miRNA-specific stem loop primer (see S3 Table), followed by (2) quantitative PCR using both a miRNA-specific forward primer and an universal reverse primer [41] (see S3 Table).

## Chromatin immunoprecipitation (ChIP) and qPCR

Immunoprecipitation assay was performed with specific anti-dMyc (P4C4-B10, DSHB) and anti-GFP (12A6, DSHB) antibodies on L3 control ($w^{1118}$) larvae following the modENCODE protocol [42,43]. The specific immunoprecipitated DNA was detected by quantitative PCR using primers listed in S3 Table.

## ChIP-seq data analysis

modENCODE ChIP-seq database (ENCSR191VCQ) from third instar larvae was analyzed to identify putative dMyc-regulated miRNA genes considering peaks located closer than 5kb to a TSS (Transcription Start Site) in replicated experiments and above optimal IDR (Irreproducibility Discovery Rate) threshold (https://epic.gs.washington.edu/modERN/). dMyc binding to the genomic region of 112 miRNA genes was also predicted through TransmiR ChIP-seq databases (ERX242709 and SRX160967; https://www.cuilab.cn/transmir). R (version 4.0.2) and Bioconductor (version 3.16) were used to define miRNAs' genomic locations and to calculate distances to the nearest dMyc peak (TxDB: 10.18129/B9.bioc.TxDb.Dmelanogaster.UCSC.dm6.ensGene; [44]).

## Bioinformatic analysis

The FIMO package from the MEME suite (https://meme-suite.org/meme/tools/fimo) was used to analyze the presence of E-box sequences on DNA regions overlapping with Myc peaks. The following Myc motifs obtained from JASPAR database [45] were used: (1) canonical E-box sequence (CACGTG) corresponding to human Myc (MA0147.3), human Myc:Max heterodimer (MA0059.1), and human MycN (MA0104.4); and (2) non-canonical E-box sequences (CACATG) corresponding to human Max binding site (MA0058.2). An unbiased *de novo* motif discovery using MEME (https://meme-suite.org/meme/tools/meme) was also performed. DNA sequences from the most significant Myc peaks from modENCODE datasets (signal enrichment above 95% corresponding to 800 peaks) were used as input. An enriched motif, named MEME2, was identified and used for the analysis (S2 Fig).

## Quantification and statistical analysis

For starvation sensitivity assays statistics were performed using GraphPad Prism6, which uses the Kaplan-Meier estimator to calculate survival fractions as well as median and maximum survival values. Curves were compared using the log-rank (Mantel-Cox) test. The two-tailed p value indicates the value of the difference between the two entire survival distributions at comparison.

Graphpad Prism6 was used for statistical analysis and graphical representations based on three or more replicates for each experiment. All significance tests were carried out with unpaired two tailed Student's t tests. Significance P values: *p < 0.05, **p < 0.01, ***p < 0.001, ****p < 0.0001, $^{ns}$p > 0.05.

Images were acquired on a Leica SP8 inverted confocal microscope and analyzed and processed using Fiji [46] and Adobe Photoshop. Tissue orientation and/or position was adjusted in the field of view for images presented. No relevant information was affected.

## Supporting information

**S1 Fig. Related to Fig 1.** (A) qRT-PCR showing *myc* mRNA levels in control ($w^{1118}$) and *tub*>dMyc larvae. (B) qRT-PCR showing pri-miR expression in control ($w^{1118}$) and *tub*>d-Myc larvae. Results are expressed as fold induction with respect to control animals. (C) qRT-PCR showing that *actin* mRNA levels remained unaffected in GSG$^{162}$>*myc*$^{RNAi}$ larvae with either RU486 or vehicle treatment. The results are presented as CT (cycle threshold) values for each condition. Mean ± SEM. Unpaired two-tailed t-tests: * p<0.05; ** p<0.01; ***p<0.001; ns: not significant.
(TIF)

**S2 Fig. Related to Fig 2.** (A-D) Canonical (A-C) and non-canonical (D) Myc binding E-box sequences. (E) An enriched, putative Myc-binding motif identified by an unbiased *de novo* motif discovery using MEME (see Materials and methods for details).
(DOCX)

**S3 Fig. Related to Fig 4.** (A) Wing discs carrying the p53-sensor and expressing *dcr-1*$^{RNAi}$ under the control of the *ptc*-Gal4 driver (marked by the expression of RFP, in red) stained to visualize GFP (in green or white). (B) Wing discs expressing *myc*$^{RNAi}$ under the control of the *ptc*-Gal4 driver (marked by the expression of RFP, in red) and stained to visualize Dmp53 protein expression (in green or white). (C) Wing discs expressing *myc*$^{RNAi}$ under the control of the *ci*-Gal4 driver and stained to visualize dMyc protein expression (in red or white). (D-F) Wing discs carrying the indicated miR-sensors and stained to visualize GFP (in green or white). Red arrowheads depict the anterior-posterior (A-P) boundary. A: anterior; P: posterior.
(TIF)

**S4 Fig. Related to Fig 5.** (A) Fat body cells labeled to visualize p53$^{\Delta305}$-sensor (in green or white) from starved larvae expressing dMyc (marked by the expression of RFP, in red or white). Scale bars, 20 μm. (B) qRT-PCR showing *dcr1* transcript levels in the FB of larvae from the indicated genotypes subjected to well fed (WF) or starved (STV) conditions. Results are expressed as fold induction with respect to control animals. Mean ± SEM. Unpaired two-tailed t-tests: ns: not significant.
(TIF)

**S5 Fig. Related to Fig 6.** (A-C) Survival rates to nutrient deprivation of adult flies of the indicated genotypes compared to control flies subjected to the same procedure. (A) The GSG132-Gal4 line was utilized to express dMyc or miR-305 specifically in the adult fat body. Newly eclosed adults of each genotype were transferred to food supplemented with 50 μg/ml of RU486 (Sigma). After 5 to 7 days, flies were transferred to vials containing 2% agar in PBS along with 50 μg/ml of RU486. (B) The Cg-Gal4, tub-Gal80ts line was used to express Dmp53 in the adult fat body. Newly eclosed adults of each genotype, grown at 18˚C, were switched to 29˚C. After 5 to 7 days at 29˚C, flies were transferred to vials containing 2% agar in PBS. See S2 Table for n, p-value, median, and maximum survival values. Error bars represent SEM.
(TIF)

**S1 Table. Table compiling Drosophila miRNA genes, genomic location, closest dMyc-binding site, and E-box sequences considering Modencode and TransmiR databases.**
(XLSX)

**S2 Table. Table compiling the number of individuals (n), p-values according to the Mantel-Cox test, median and maximum survival values (h) corresponding to the different**

**genotypes analyzed in Fig 6.**
(XLSX)

**S3 Table. List of primers used in this study.**
(XLSX)

## Acknowledgments

We thank Marco Milan, Vienna Drosophila RNAi Center, Drosophila Bloomington Stock Center, and the Developmental Studies Hybridoma Bank for flies and antibodies.

## Author Contributions

**Conceptualization:** María P. Gervé, Juan A. Sánchez, Andrés Dekanty.

**Funding acquisition:** Andrés Dekanty.

**Investigation:** María P. Gervé, Juan A. Sánchez, María C. Ingaramo.

**Project administration:** Andrés Dekanty.

**Supervision:** Andrés Dekanty.

**Visualization:** Andrés Dekanty.

**Writing – original draft:** Andrés Dekanty.

**Writing – review & editing:** María C. Ingaramo, Andrés Dekanty.

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
