## [Decision Letter · Decision Letter 0]

10 May 2023

Dear Dr Dekanty,

Thank you very much for submitting your Research Article entitled 'Myc-regulated miRNAs modulate p53 expression and impact animal survival under nutrient deprivation' to PLOS Genetics.

The manuscript was fully evaluated at the editorial level and by independent peer reviewers. The reviewers appreciated the attention to an important problem, but raised some substantial concerns about the current manuscript. Based on the reviews, we will not be able to accept this version of the manuscript, but we would be willing to review a much-revised version. We cannot, of course, promise publication at that time.

As you will see from their feedback, two reviewers were in principle favorable for publication, but requested several revisions, which should be addressed in full. One reviewer (#3) raised concerns regarding the conceptual novelty of the manuscript, in light of an earlier study (Barrio et al.) addressing the role of TOR-miR305-dp53 pathway in the fat body to promote survival under nutrient stress. In your response, please address this critique and revise your manuscript to include more specific description of the earlier findings and the added conceptual insight of your current study. Moreover, Reviewer 3 made specific suggestions to each figure, which should be carefully regarded and addressed.

If you decide to revise the manuscript for further consideration at PLOS Genetics, please aim to resubmit within the next 60 days, unless it will take extra time to address the concerns of the reviewers, in which case we would appreciate an expected resubmission date by email to plosgenetics@plos.org.

We are sorry that we cannot be more positive about your manuscript at this stage. Please do not hesitate to contact us if you have any concerns or questions.

Yours sincerely,

Ville Hietakangas

Academic Editor

PLOS Genetics

Gregory Barsh

Editor-in-Chief

PLOS Genetics

Reviewer's Responses to Questions

**Comments to the Authors:**

Reviewer #1: In this paper, Gerve et al describe a role for Myc as a regulator of miRNA expression and processing. They re-analyzed published ChIP seq data sets and found that Myc can directly bind to miRNA promoter regions and an also control the expression of many miRNAs. They also show that one Myc-regulated miRNA (miR305) is important in mediating the effects of adipose Myc on starvation survival.

Overall, this was a very good paper. The experiments were all well-performed and the interpretations and conclusions were supported by the findings. This study extends our appreciation of Myc target genes and how Myc controls gene expression. In this context, the work will be of interest to a broad array of researchers interested in Myc function in different systems (not just Drosophila).

These are some minor issues that the authors may choose to address before publication.

Fig 4: The effects of Myc overexpression/knockdown on the Myc sensor seem quite weak. In fig 4D also looks like the down regulation of the myc-sensor is non-autonomous – the stripe of decreased expression of the sensor seems wider than the dpp>RFP expression domain. Is this case? If so, it might be worth commenting on. Or is there a lower expression level in this domain in the absence of myc expression. For this figure, it would be good to add a control image for each sensor showing the expression pattern of the sensor in the absence of Myc expression.

Fig 5. In Fig 5C, the authors use the tub>Myc system to see if Myc overexpression can reverse the starvation-induced decrease in dcr1. They saw ~3fold upregulation of Myc mRNA but no reversal of the starvation mediated decreased. However, starvation can posttranscriptionally suppress Myc levels and so I wonder whether there is sufficient upregulate of Myc protein levels with the tub>Myc system. Would it be worth trying a stronger Myc expression, maybe using heatshock induction with the hsflp-out system (Fig 5E, F).

Reviewer #2: Revision report of Gervé et al. Myc-regulated miRNAs modulate p53 expression and impact animal survival under nutrient deprivation

The authors show that dMyc regulates a large share of miRNA genes as well as the miRNA machinery in the fly larva. They then use a miRNA sensor to show the regulation of dp53 by dMyc and miR305 in vivo, in wing disc and in fat body. Finally, the authors demonstrate a physiological relevance for the dMyc-miR305-dp53 pathway under nutrient deprivation.

Overall, the manuscript is well written, the methods and the data are clearly presented and for the most part the data is supporting the conclusions. My main concern is that it is unclear to what extent this study expands the already known mechanism published by the corresponding author previously (https://doi.org/10.1016/j.celrep.2014.06.020). In Barrio et al. (Cell Reports 2014) it was shown that TOR-miR305-dp53 pathway in the fat body promote survival under nutrient stress. While the finding that Myc is a major regulator of miRNAs is potentially interesting, it is also well known that Myc is acting downstream of TOR signaling. Hence, the finding that Myc is a mediator in their previously published mechanism is, to some extent, expected.

Specific comments figure by figure:

Fig 1.

Please indicate the applied statistical test used in the Figure legend (applies to all figure legends).

Since dMyc is a major regulator of cellular growth, its downregulation is expected to reduce the level of overall transcription as well. To make the results more convincing, could the authors provide the data of their reference gene actin, to show its expression has remained constant in their experiments. This data can be provided in the supplement.

The authors conclude “Together, these findings indicate that dMyc regulates the expression of a large number of miRNAs in Drosophila”, yet the number of affected pri-miRNAs between dMyc LOF and GOF are quite different (10 in LOF vs. 3 in GOF). Could the authors clarify in the text why such difference exists?

Figure 1 title states that “Myc induces widespread expression of miRNA genes in Drosophila”. Could the authors rephrase the title since the word “induces” refers to Myc upregulating the miRNA levels which is not shown here.

Fig 2.

The conclusions are justified by the provided data.

Fig 3.

To make the regulatory function of dMyc to the elements of the miRNA machinery more clear, could the authors also show the dMyc GOF data for the ago1, dcr-1 and drosha expression?

Fig 4.

It is not completely clear why the results from the meip26-sensor and myc-sensor are shown. Could the authors clarify this in the text?

To make the results more convincing, a quantification of the GFP signal along the wing disc A-P axis is required.

In (A), the RFP signal is missing from the figure.

Fig 5.

In S1 pri-mir305 was not affected by dMyc overexpression, yet in Fig5 it is affected by dMyc overexpression. The reason might be that the measurements are from different tissues. Could the authors clarify this in the text?

Could the authors change ‘well fed’ into ‘fed’ and STV into starv? This would make the interpretation of the figure easier for the reader.

In panel C, the authors show that dcr-1 and drosha levels were not rescued by the expression of dMyc in starved versus fed larvae. Based on this a conclusion of an indirect regulation was made. However, in 3B, by ChIP-qPCR, the authors show the direct binding of dMyc into the drosha promoter. Could the authors clarify this discrepancy?

Could the authors clarify the details of the clone technique used in Fig5 E & F in the figure legend? This would make it easier for the reader to understand the nature of the experiment. Please also clarify the used ‘AFGal4’ driver.

Fig 6.

Is the dMyc and miR305 knockdowns present throughout the fly development, and if yes, what are the consequences of dMyc and mir305 manipulation during development to the adult flies? It would be important to understand if the adult starvation sensitivity/insensitivity is due to developmental defects or specific to the adult stage.

The results from overexpressing p53 in the fat body are not shown. According to the model, overexpressing p53 in the fat body should increase starvation sensitivity. To make the conclusions solid, this result should be provided. Alternatively, a reason for not conducting this experiment should be provided.

In panel E, the Lsp2>+ is not shown. Also the data is only shown for male flies. To make the conclusion of the myc-p53 interaction in starvation sensitivity convincing, the authors should provide the Lsp2>+ control, and the female data (even if negative). Or at least provide a reason why the female data was omitted.

Please indicate the fly gender also in the figure panels.

In line 182 is mentioned Fig 6F, but this panel is not included in figure 6.

A model figure explaining the dMyc-mir305-p53 pathway would be helpful for the reader.

Reviewer #3: In this MS, Gervé and cols analyze the contributions of the transcription factor Myc to the expression of miRNAs. They use Drosophila as a model to perform this analysis. They find that changes in Myc activity result in profound changes in miRNA expression. They also show that central elements controlling miRNA biogenesis are also altered upon changes in Myc activity. At the end of the MS, the authors focus their attention in miR-305 and its regulation by Myc in a context of starvation resistance.

The MS is, in general, and explores a mechanism of central interest in the context of animal development and homeostasis. However, several issues need to be addressed befor it is ready for publication.

See specific comments below:

In the first part of the MS, the authors analyze the expression of a selected group of miRNAs in a context of Myc downregulation. However, the authors do not explain the criteria used to select those miRNAs. An explanation should be included.

The authors analyse the changes in the expression in pre-miRNAs and pri-miRNAs. Although pri-miRNAs are explained in the introduction, pe-miRNAs are not introduced. To facilitate the understanding by non-specialists, a proper introduction of these terms (pri-, pre-, and mature miRNAs) will be required.

Fig 1C, D, shows changes in a reduced number of miRNAs. Why only those? There is lack of cohesion between the results shown in B and C,D . I would suggest covering the analysis of pri-, pre-, and mature for the same set of miRNAs. This would provide a more complete view of this process.

Previous studies by the Milán group showed that dMyc regulate bantam (PMID: 18451803) are consistent with this study. Those results should be mentioned here.

The second miRNA shown is 219 (in C) and 279 (in D). Is that a typo or is it correct? The other 3 miRNAs shown are the same in both graphs.

I would also suggest merging Fig 1 and Fig S1. The results shown are complementary and having them together will provide a more complete picture of the regulatory roles of dMyc.

Fig 2 shows that Myc bind the promotor of numerous miRNAs. If this specific for miRNAs or a similar binding can be observed in other classes of genes?

Fig 3. Does Myc over-expression increase those transcripts? This complementary analysis might help to understand the results shown in Fig S1.

In line 130, the authors mentioned: “The Drosophila ortholog of mammalian p53 (Dmp53) expression has previously been shown to be under control of miRNAs in both the wing imaginal disc and the fat body [28]. Interestingly, the levels of 3 miRNAs (miR-283, miR-219, miR-305) previously shown to regulate Dmp53 expression in the wing primordium [28] were reduced upon mycRNAi expression (Fig. 1B) and showed dMyc binding to its genomic region (Fig. 2D, Table S1).” This is followed by “We then used the larval wing primordium to characterize dMyc contribution in miRNA biogenesis.” I can´t find a logic connection between these two parts.

The authors showed that Myc controls miRNA expression and biogenesis. The results shown in Fig 4 bring them to conclude that Myc regulates the miRNA machinery. However, the changes shown in this Fig could partially result from changes in the expression of miRNAs and not only due to the biogenesis machinery. I would suggest the authors to consider that option.

**Have all data underlying the figures and results presented in the manuscript been provided?**

Reviewer #1: Yes

Reviewer #2: Yes

Reviewer #3: Yes

PLOS authors have the option to publish the peer review history of their article (what does this mean?). If published, this will include your full peer review and any attached files.

Reviewer #1: No

Reviewer #2: No

Reviewer #3: No

---

## [Decision Letter · Decision Letter 1]

16 Aug 2023

Dear Dr Dekanty,

We are pleased to inform you that your manuscript entitled "Myc-regulated miRNAs modulate p53 expression and impact animal survival under nutrient deprivation" has been editorially accepted for publication in PLOS Genetics. Congratulations!

Yours sincerely,

Ville Hietakangas

Academic Editor

PLOS Genetics

Gregory Barsh

Editor-in-Chief

PLOS Genetics

Comments from the reviewers (if applicable):

Reviewer's Responses to Questions

**Comments to the Authors:**

Reviewer #1: The revised manuscript addresses all my previous concerns and is suitable for publication.

Congratulations to the authors on a very nice story and paper.

Reviewer #2: The authors have addressed all my suggestions and concerns in the revised version of the manuscript.

Reviewer #3: The authors have addressed all my concerns and the manuscript is now ready for publication.

**Have all data underlying the figures and results presented in the manuscript been provided?**

Reviewer #1: Yes

Reviewer #2: Yes

Reviewer #3: Yes

PLOS authors have the option to publish the peer review history of their article (what does this mean?). If published, this will include your full peer review and any attached files.

Reviewer #1: No

Reviewer #2: No

Reviewer #3: No

**Data Deposition**

http://datadryad.org/submit?journalID=pgenetics&manu=PGENETICS-D-23-00345R1

**Press Queries**

---

## [Editor Report · Acceptance letter]

24 Aug 2023

PGENETICS-D-23-00345R1 

Myc-regulated miRNAs modulate p53 expression and impact animal survival under nutrient deprivation 

Dear Dr Dekanty, 

We are pleased to inform you that your manuscript entitled "Myc-regulated miRNAs modulate p53 expression and impact animal survival under nutrient deprivation" has been formally accepted for publication in PLOS Genetics! Your manuscript is now with our production department and you will be notified of the publication date in due course.

With kind regards,

Zsofia Freund

PLOS Genetics

On behalf of:
